# Body Composition and Metabolic Dysfunction Really Matter for the Achievement of Better Outcomes in High-Grade Serous Ovarian Cancer

**DOI:** 10.3390/cancers15041156

**Published:** 2023-02-10

**Authors:** Mauricio A. Cuello, Fernán Gómez, Ignacio Wichmann, Felipe Suárez, Sumie Kato, Elisa Orlandini, Jorge Brañes, Carolina Ibañez

**Affiliations:** 1Department Gynecology, School of Medicine, Pontificia Universidad Católica de Chile, Santiago 833150, Chile; 2Advanced Center for Chronic Diseases (ACCDiS), Faculty of Medicine, Pontificia Universidad Católica de Chile (PUC), Santiago 833150, Chile; 3Department of Obstetrics, Division of Obstetrics and Gynecology, Faculty of Medicine, Pontificia Universidad Católica de Chile (PUC), Santiago 833150, Chile; 4Division of Oncology, Department of Medicine, Stanford University School of Medicine, Stanford, CA 94305, USA; 5Department of Hematology & Oncology, School of Medicine, Pontificia Universidad Católica de Chile, Santiago 833150, Chile

**Keywords:** epithelial ovarian cancer, body composition, inflammation, immunity, treatment outcomes

## Abstract

**Simple Summary:**

Current evidence supports a negative impact of obesity-associated metabolic dysfunction in several cancers. However, the evidence is still controversial regarding high-grade serous ovarian cancer (HGSOC). In this study, we demonstrated that body composition, particularly the presence of high visceral adiposity (with or without sarcopenia) estimated by aCT scan, is associated with worse survival in HGSOC. As a molecular proxy to CT-scan-based assessment of nutritional status and to identify putative biomarkers of metabolic disorders, we evaluated the expression levels of a set of 425 obesity- and lipid-metabolism-disorder-related genes across 273 tumor samples. We identified two obesity- and lipid-metabolism-related clusters with marked differences in survival and that were associated with molecular features predictive of immune checkpoint blocker response. Finally, we assessed the impact of nutritional/pharmacologic interventions affecting body composition/lipid metabolism on patient survival. We observed that the reduction of visceral adiposity, the increase of muscle mass, and the use of metformin/statins improve survival.

**Abstract:**

Although obesity-associated metabolic disorders have a negative impact on various cancers, such evidence remains controversial for ovarian cancer. Here, we aimed to evaluate the impact of body composition (BC) and metabolism disorders on outcomes in high-grade serous ovarian cancer (HGSOC). Methods: We analyzed clinical/genomic data from two cohorts (PUC n = 123/TCGA-OV n = 415). BC was estimated using the measurement of adiposity/muscle mass by a CT scan. A list of 425 genes linked to obesity/lipid metabolism was used to cluster patients using non-negative matrix factorization. Differential expression, gene set enrichment analyses, and Ecotyper were performed. Survival curves and Cox-regression models were also built-up. Results: We identified four BC types and two clusters that, unlike BMI, effectively correlate with survival. High adiposity and sarcopenia were associated with worse outcomes. We also found that recovery of a normal BC and drug interventions to correct metabolism disorders had a positive impact on outcomes. Additionally, we showed that immune-cell-depleted microenvironments predominate in HGSOC, which was more evident among the BC types and the obesity/lipid metabolism cluster with worse prognosis. Conclusions: We have demonstrated the relevance of BC and metabolism disorders as determinants of outcomes in HGSOC. We have shone a spotlight on the relevance of incorporating corrective measures addressing these disorders to obtain better results.

## 1. Introduction

Despite recent advances in understanding the biology of high-grade serous ovarian cancer (HGSOC) and the addition of new therapeutic avenues based on molecular targets (i.e., anti-angiogenics, poly (ADP-ribose) polymerase inhibitors (PARPi), or immune checkpoint blockers [ICB]), survival remains very poor [1,2]. Currently, the best chance of long-term survival is achieved when the tumor burden is optimally debulked by upfront surgery (or after three cycles of neoadjuvant chemotherapy), and any residual microscopic disease is amenable to available adjuvant options [1,2,3]. However, the most frequent scenario will be one of tumor recurrence and subsequent death. Mainly because of cost issues, long-term toxicity, and therapy resistance, maintenance therapies do not usually extend beyond two years [1,4]. Therefore, most efforts focus on leveraging molecular information to identify the tumor’s “Achilles heel” to offer a suitable alternative for such an individual case—perhaps as the ‘only shot’ available for a disease still considered as non-curable. The emphasis is on offering a cost-effective therapy, with a low complication rate, that is both affordable and well-tolerated by the patient [2].

After primary treatment is completed, patient surveillance includes management of the side effects of therapy and timely identification of predictors of recurrence and susceptibility to future salvage schemes in those places where such options exist. This approach justifies the search for mutations of somatic/germinal genetic defects (homologous recombination deficiency [HRD] or BRCA1/2 mutations) that predict sensitivity to novel targeted therapies [2]. Unfortunately, neither physicians nor patients, independently of the socioeconomic and cultural setting, routinely emphasize non-genetic extra tumoral host factors that may determine tumor biology, early recurrence, or future response to these salvage options.

Recently, in addition to chronological age and baseline performance status, modifiable factors such as obesity and certain co-morbidities (e.g., diabetes and hypercholesterolemia) have become more relevant in terms of defining the course of the disease, therapeutic options, and the chance of response in several cancers [5]. Despite this evidence, the prognostic role of obesity in HGSOC is still contradictory [6,7]. Part of the controversy arises from the inclusion of diverse histological classes and stages in the previous series. In addition, most efforts to assess the impact of obesity on disease development rely on the use of the body mass index (BMI) as its defining variable. The evidence arising from studies of metabolic syndrome and the increasing incidence of non-alcoholic fatty liver disease (NAFLD) in patients with normal BMI has led to rethinking the definition of obesity and diagnostic methods and to the establishment of new conditions such “healthy obese” and “lean metabolically sick individuals” with different cardiovascular and cancer risks compared with those defined by BMI as “morbidly obese” and “lean individuals” [8]. Therefore, alternative approaches including the use of body composition, visceral adiposity, muscle mass and/or metabolic disorders, should be considered as part of the concerted efforts to refine this definition.

Here, we evaluate whether changes in the status of different predictive factors, measured at diagnosis and after completion of first-line treatment, impact on progression-free (PFS) and overall survival (OS) in HGSOC. Among them, we included (i) reduction of tumor burden (optimal debulking), (ii) type of response (complete, partial, and no response), (iii) CA-125 levels, and (iv) inflammatory status (i.e., ANC/ALC, LDH levels, and the systemic immune-inflammatory index [SIII]) [9,10,11,12,13]. We also evaluated the impact of (a) body composition, using alternative measures to BMI encompassing whole-body adipose tissue (WBAT) and muscle mass; and (b) metabolic disorders (i.e., obesity and lipid metabolism-related gene expression profiles, liver steatosis, and blood cholesterol levels) on the risk of recurrence and the survival outcomes of HGSOC patients.

## 2. Materials and Methods

### 2.1. Patient Data Collection

This study was IRB-approved (ID 190408002, 5 July 2020). Electronic records from patients with confirmed epithelial ovarian cancer who were treated at our institution (PUC) between 2004 and 2017 were reviewed. Only stage III and IV HGSOC with at least six months of follow-up were included in the study. All of the cases included in the survival analyses had information on histological confirmation of diagnosis, age, adequate staging, intent of surgery, chemotherapy schemes, and tumor burden/residual disease after planned treatment completion. For those cases undergoing interval debulking surgery (IDS), we registered if the surgery was carried out in timely fashion as planned (after the third or fourth cycle of chemotherapy). All of the cases with ‘incomplete or unclear’ information on these relevant factors were excluded. To guarantee maturity of the survival analyses, we collected data until at least 50% of the patients had experienced death by any cause. Date and cause of death were verified by accessing the Chilean Civil Registry database. The final dataset included the following clinical variables: age, height (cm), weight (kg), body mass index (BMI), FIGO stage, type of surgery (upfront, interval, or no surgery), debulking level (microscopic or any size macroscopic residual disease), type of response (complete, partial/stable, or progressive disease), type and number of chemotherapy schemes, the absolute neutrophile count (ANC) to absolute lymphocyte count (ALC) ratio (ANC:ALC), absolute platelet count per ml, the calculated systemic immune-inflammatory index (platelets x ANC/ALC), lactate dehydrogenase (LDH), albumin, and cholesterol levels.

In addition, we downloaded publicly available raw counts and upper-quartile normalized fragments per kilo base per million mapped (FPKM-UQ) RNA-seq expression and clinical data from Genomic Data Commons through the TCGA-biolinks R package [14]. As with the PUC cohort, the same inclusion criteria were used. TCGA-OV patients with ‘incomplete or unclear’ clinical information were also excluded from the survival analyses. Log_2_-transformed FPKM-UQ were used to perform further transcriptome-based analyses related to the impact of body composition on progression-free (PFS) and overall survival (OS).

### 2.2. Genetic and Clinical Assessment of Nutritional Status and Lipid-Metabolism-Related Disorders

For the RNA-seq analyses, a curated list of 425 genes linked to obesity and lipid metabolism disorders was aggregated from different sources [15,16]. Unsupervised clustering using non-negative matrix factorization (NMF) was performed on the TCGA-OV dataset, based on the log_2_-FPKM levels of these 415 genes [17]. The cophenetic correlation coefficient and the average silhouette width calculation were used to determine the most robust clusters. Differential expression analyses were carried out using raw counts and the comparative marker selection module (genepattern, Broad Institute, Cambridge, MA, USA [18]) to calculate the significant differences in gene expression between classes. Gene set enrichment analysis (GSEA), gene ontology (GO), hallmarks, KEGG pathways enrichment, cell states, and analysis of cellular communities in TCGA-OV patients were carried out using GSEA 4.3.2 (Broad Institute, Cambridge, MA, USA [19]), ShinyGO 0.76.2 (South Dakota State University, Brookings, SD, USA [20]), and Ecotyper (Stanford University, Stanford, CA, USA [21]). Available data on molecular classification, microenvironmental subtypes, tumor mutational burden (TMB), stemness index, and ESTIMATE scores (https://bioinformatics.mdanderson.org/estimate/ accessed on 10 December 2022) were compiled from different sources [21,22,23,24,25,26].

For clinical analyses, CT scans were downloaded from either our institutional server or the Cancer Imaging Archive (TCIA, NCI, USA). To assess nutritional status and body composition, the visceral adipose tissue (VAT), subcutaneous adipose tissue (SCAT), the psoas muscles, and the vertebral body areas were measured using a single-slice CT scan located between the 4th and 5th lumbar vertebrae. Next, VAT/SCAT ratio, VAT/TAT ratio (VAT/VAT + SCAT), and Psoas to L4 vertebral body index (PLVI) were calculated to estimate visceral obesity and central sarcopenia [27]. Visceral obesity was defined by either visceral adipose tissue area ≥100 cm^2^, VAT/SCAT ≥ 0.4, or VAT/TAT ≥ 0.285 [28]. A cut-off value of 0.45 for PLVI defined central sarcopenia (the lowest quartile of the sample). Whole-body adipose tissue mass was additionally estimated by the Lacoste’s formula (0.0677 * AT_L4-L5_ + 2.5177) [29]. Liver steatosis, a marker of metabolic dysfunction and lipid metabolism disorders, was estimated by using the CT scan Hounsfield Unit (HU) difference between liver and spleen (CT_L-S_) at a cut-off value of −20 [30]. The image analyses were carried out using OsiriX MD v.12.5.3.

### 2.3. Co-Morbidities and mFI-5

Co-morbidities such as hypertension, diabetes mellitus, heart failure (CHF), chronic obstructive pulmonary disease, and non-independent functional status were registered to calculate the modified frailty index 5 (mFI-5) [31,32,33]. An mFI-5 ≥ 1 was assigned as higher risk.

### 2.4. Statistical Analyses

JMP16.2.0 (SAS Institute Inc., Cary, NC, USA) was used for the data analyses. Group comparisons and continuous or categorical variable association analyses were performed as required by the Student’s *t*-test, Chi-squared test, Mann–Whitney U-test, or logistic regression. A *p*-value < 0.05 was considered statistically significant. Survival curves were generated by using the Kaplan–Meier method and analyzed by Wilcoxon and log-rank tests. Univariate Cox proportional hazard regression analyses were carried out for all of the variables. Multivariate Cox proportional hazard regression models were carried out using a stepwise method to assess the effects of different risk factors on survival. To select variables to be included in the modeling, we used a stopping rule or selection criteria based on the *p*-value of 0.2 [34]. We also added variables based on clinical relevance reported in the literature, independently of the *p*-value found [16,35,36,37].

## 3. Results

### 3.1. Cohort Characteristics and Clinical Parameters Associated with Disease Recurrence and Survival

A total of 123 and 415 stage III and IV HGSOCs from our institution (PUC) and the TCGA-OV cohort were included in the present study, respectively. Twenty-two cases in the PUC cohort and 102 cases in the TCGA-OV cohorts were excluded for lacking relevant clinical information (e.g., histology, adequate staging, type of surgery, debulking status, and adequate follow-up) or excessive and unjustified delay in accomplishing planned treatment (for the PUC cohort). As shown in Table 1, these were two globally comparable cohorts with adequate follow-up. In both cohorts, the major determinants for recurrence and overall survival were achieving debulking at the microscopic level and complete response, regardless of the therapeutic sequence (primary debulking surgery [PDS] followed by chemotherapy or interval debulking surgery [IDS]). Regarding inflammatory (e.g., LDH levels, ANC/ALC, and SIII) and nutritional (i.e., albumin) parameters, they tended to normalize in most of our cohort (Appendix A). In a univariate analysis, such normalization correlated with overall survival (Appendix A). It should also be noted that non-modifiable determinants such as chronological age (≥70 years of age) and the frailty index (mFI-5 ≥ 1) had a negative impact on overall survival.

### 3.2. Body Composition, and Not BMI, Is Associated with Patient Outcomes

Patient BMI remained unchanged in our cohort after six months, by the end of the first-line treatment (Figure 1A). The presence of obesity or being overweight did not determine a higher risk of recurrence or lower PFS or OS (Figure 1B,C). When estimating visceral adiposity (VAT), whole body adipose tissue (WBAT), and central muscle mass (psoas to lumbar vertebral body index [PLVI]) by CT scan, we observed that there were differences in the distribution of adipose tissue and not in PLVI between women who were obese, overweight, and a normal weight as defined by their BMI. Of note, a low PLVI (within the central sarcopenia range) was found in most of the cases in our cohort and independently of BMI status (see Figure 1D–F). Additionally, we observed that some women with normal a BMI presented high adiposity as defined by WBAT, while some obese and overweight women presented normal adiposity (Figure 1G). More importantly, significant changes in WBAT, VAT, and VAT/TAT were observed post-treatment and persisted at the 1-year follow-up (Figure 1H and Figure 2A,B), suggesting that the PLVI does not correct even by one-year post-treatment for most of the cases (Figure 2C).

By combining the VAT or WBAT with PLVI estimates, we were able to establish four types of body composition (BC) in our patients. As shown in Figure 2D,E, at least three out of four BC types were present in patients traditionally defined as obese, overweight, or normal weight by BMI. Likewise, it is important to highlight that regardless of how we constructed the estimate, BC types with high total or visceral adiposity and normal PLVI were predominant. We also observed a BC type where high adiposity coexisted with central sarcopenia independently of BMI (Figure 2D,E).

In terms of survival, and unlike what was observed with BMI, both PFS and OS were significantly different between BC types. Patients with normal visceral adiposity and normal muscle mass had significantly better survival compared to any of the other types. The worst prognosis was observed in the presence of high adiposity and central sarcopenia (Figure 2F,G). A similar distribution of BC types was found in the TCGA compared to the PUC cohort (Appendix A). In addition to body composition, we estimated the prevalence of hepatic steatosis, a marker of metabolic dysfunction (hepatic lipid metabolism disorder) in the different BC types. As expected, there was a trend towards a higher proportion of hepatic steatosis in patients with high adiposity (30.4% vs. 18.2%, *p* < 0.001) independent of BMI status in both cohorts (Table 1 and Appendix A).

Since the two cohorts were comparable, our next step was to analyze the pooled data (using only cases where a surgical intent of debulking was made, either upfront or as interval surgery) to establish a formula maybe applicable to other cohorts. Thus, we confirmed a cut-off point already validated by others for the VAT/SCAT (0.4) and we established a cut-off value for the PLVI based on the lowest quartile of the sample, 0.45. The aggregated set included 201 cases. Both the VAT/SCAT and PLVI cut-off points were associated with lower overall survival in univariate analysis (Figure 3A,B). In addition, the BC types defined either using VAT/SCAT or WBAT with such PLVI cut-offs were associated with patient survival (Figure 3C,D). Regarding survival differences among different BC subtypes, the biggest difference was observed between the BC subtype with high adiposity (visceral or total body) and central sarcopenia and the BC subtype with normal adiposity and normal muscle mass, either estimated by VAT/SCAT or WBAT and PLVI, with the high VAT/SCAT or high WBAT and low PLVI BC subtype having the worse OS. Moreover, the defined BC types constituted significant prognostic factors independently of achieving a complete response with first-line treatment, age, or their mFI-5 (Figure 3E,F).

### 3.3. Genes Related to Obesity and Lipid Metabolism Distinguish Two Clusters of Patients with Marked Differences in Survival in the TCGA-OV Cohort

As a molecular proxy to CT-scan-based assessment of nutritional status and to identify putative biomarkers of metabolic disorders, we evaluated the expression levels of a set of in-house precompiled 425 obesity- and lipid-metabolism-disorder-related genes (Appendix A) across 273 tumor samples from the TCGA-OV cohort. Using non-negative matrix factorization, we identified two clusters (from here on referred to as obesity/lipid metabolism type I and II) with distinct overall survival (Figure 4A). A trend towards better PFS in the same cluster was also observed (obesity/lipid metabolism type II). These two clusters differentially expressed sets of genes associated with different biological processes, hallmarks, and signaling pathways (see Appendix A). Briefly, to exemplify, in the cluster with the best prognosis (obesity/lipid metabolism type II), we observed higher expression of genes related to processes of intracellular localization of proteins and macromolecules, while in the one with the worst prognosis (obesity/lipid metabolism type I), genes related to cell-to-cell signaling and neuron differentiation predominated. Regarding hallmarks and signaling pathways, metabolic pathways were more highly expressed in the cluster with the best prognosis (e.g., adipogenesis and fatty acid metabolism), while pancreatic ß-cell and neuroactive ligand–receptor interaction pathways predominated in the cluster with the worst prognosis (Appendix A).

### 3.4. Obesity- and Lipid-Metabolism-Related Clusters Associate with Molecular Features Predictive of ICB Response

We compared established patterns and scores that correlate with prognosis and ICB response between the obesity/lipid metabolism type I and obesity/lipid metabolism type II clusters, namely: stemness index, microenvironmental subtype, tumor mutation burden (TMB), and ESTIMATE score [22,26,38,39,40]. Consistent with worse clinical outcomes, patients from cluster type I showed worst parameters compared to cluster type II. Moreover, the type I samples had a significantly lower proportion of the immunoreactive molecular subtype from TCGA-OV, a lower proportion of the microenvironmental subtype enriched in immune cells, lower TMB, higher stemness index, and lower ESTIMATE score (Figure 4B–G).

### 3.5. Both Obesity and Lipid Metabolism Clusters and BC Types Have Different Compositions of Immune Cell Types and States

Next, we used the EcoTyper framework that enables the identification of cellular states and communities (cancer ecotypes or CE) from bulk RNA-seq data. Different cell types interact with each other in different functional states (cell states or S) and together condition the relationship between the tumor and host in a specific ecosystem [21].

The obesity and lipid metabolism type II cluster (the cluster with the best prognosis) displayed a significantly higher proportion of CE9 and CE10 ecotypes. We also observed differences in the status of CD8, NK, plasmatic, B, dendritic, and monocyte/macrophage cells but not in CD4 or polymorphonuclear cells (Figure 4H,I, and Appendix A).

Given that the clusters were identified by gene sets related to obesity and lipid metabolism disorders and not related to sarcopenia, we decided to evaluate the impact of the association of this condition on survival already established by the clusters. We found that the presence of sarcopenia (low PLVI) made the differences in overall survival between clusters more evident, particularly for the cluster with the worst prognosis. More importantly, this association constituted a prognostic variable independent of the patient’s age, achieving complete removal of the disease and obtaining a complete response (Figure 5A and table below it).

As with the clusters, we compared CE and cell states between the BC types. Despite dealing with a smaller number of cases (54 patients), we found differences, with the CE1 ecotype predominating in the BC type with the worst prognosis (high adiposity [high WBAT] and central sarcopenia [low PLVI] type, Figure 5B). We also observed differences in plasma cell states among BC types (Figure 5C) with only two cell states (S01 and S02) found in the BC type with the worst prognosis.

### 3.6. Reduction of Visceral Adiposity, Increase of Muscle Mass, and Use of Metformin and Statins Improve Patient Survival

Finally, we assessed the impact of nutritional interventions or the addition of common medications (e.g., statins or metformin) affecting body composition and lipid metabolism on patient survival. In the PUC cohort, we observed that the improvement in VAT/SCAT and PLVI parameters was associated with reduction of visceral adiposity and recovery of muscle mass at 12 months of follow-up and improved progression-free (PFS) and overall survival (OS) (Figure 6A,B). Thus, reducing visceral adiposity and recovering muscle mass after treatment constituted an independent risk factor (Figure 6C,D, and Appendix A). The use of statins and metformin had a similar effect, showing a trend towards better survival among users of these medications (Appendix A).

## 4. Discussion

In the present study, we confirmed that most women with advanced HGSOC (67.6%) achieve a complete response (in 37.6% of them associated with complete tumor debulking achieved either primarily or after receiving neoadjuvant chemotherapy). In addition to a complete response, other factors associated with survival are the age at diagnosis, the baseline frailty of the patient (related to performance status) and, as we have shown here, the normalization of parameters related to inflammatory (e.g., LDH levels) and immune (e.g., SIII) responses at the end of first-line treatment. Enhancing the concept that ´transitioning from an inflamed to a non-inflamed patient’ besides achieving complete response constitutes a relevant factor to improve long-term outcomes [9,41].

Regarding nutritional status and metabolic disorders, we show that BMI does not change significantly after treatment in our cohort and does not correlate well with changes observed in cholesterol or albumin levels. It also does not adequately correlate with the presence of liver steatosis, whole body adiposity (high WBAT), or visceral adiposity (high VAT or VAT/SCAT) or with the presence of central sarcopenia (low PLVI) measured by a CT scan. More importantly, being a normal weight, overweight, or obese based on the BMI parameter does not constitute a predictor of progression-free or overall survival. This contrasts with the significant impact on survival of four body composition (BC) types identified using the combination of VAT or VAT/SCAT or WBAT with PLVI. Our findings confirm the coined concept of the ‘obesity paradox’ where having a BMI ≥ 30 kg/m^2^ does not correlate with the risk of chronic diseases and death from all causes during aging [42,43]. Of note is the high prevalence of central sarcopenia in the pooled cohort (27.4%), which has recently been identified as a negative risk factor in terms of survival [44]. Thus, also, the occurrence of hypercholesterolemia and liver steatosis, both markers of lipid metabolism disorders and recently raised as modulators of anti-tumor immune response as well as potential predictors of response to immunotherapy [41,45,46]. We further demonstrated that those women who maintain or achieve a ‘normal’ body composition (low WBAT or VAT or VAT/SCAT and high PLVI) at one year of follow-up indeed exhibit better progression-free and overall survival [47]. This is also the case for those who receive statins (with or without metformin) as part of the management of hypercholesterolemia, NAFLD, insulin resistance, or metabolic syndrome [48,49].

Another finding of our research is related to the expression profiles of genes related to obesity and lipid metabolism and their impact on the composition and cell states of the tumor microenvironment (TME) and on patient survival. Here, we showed that the differential expression of these genes within the tumor defines two clusters with distinct survival outcomes. These clusters also differ in the biological processes and signaling pathways that characterize them. However, more importantly, the ecosystems present in them also differ and determine different cellular interactions, particularly among those linked to the anti-tumor immune response. In particular, the cluster named here as obesity/lipid metabolism type I, in addition to exhibiting the worst prognosis, has characteristics that reflect a TME where the anti-tumor immune response is at least deficient, if not permissive. Regardless of the bioinformatic tool or score used, this cluster is characterized by a lower proportion of the immunoreactive molecular subtype, a lower relative proportion of TME defined as immune-enriched, and a higher proportion of TME defined as immunodepleted, particularly of lymphocytes. A more in-depth analysis using EcoTyper revealed that those ecosystems defined as immunoreactive, pro-inflammatory, and more sensitive to immunotherapy (C9 and C10) represent less than 20% of the ecosystems identified in this cluster [21]. When analyzing the functional state of the immune cells present in the ecosystem, we observed preponderance of cellular states that have been associated with worse prognosis in other carcinomas [21]. Of note, we reported contrasting differences between clusters in the cellular status of intra-tumoral B and plasma cells. The presence and functionality of these cell types have become relevant in determining and predicting the response to immunotherapy based on immune checkpoint blockage independent of CD8+ T-cell signals [50,51].

Finally, we must mention that regardless of the BC type, the carcinoma ecotype that predominates in advanced HGSOC is CE1. CE1-high tumors are described as lymphocyte deficient [21]. Along with this, we observed that the interactions and states of the intra-tumoral immune cells (e.g., plasma cells) vary depending on the BC type [44]. Such characteristics may partly explain the low response rate of this cancer to immunotherapy and the importance of normalizing BC (especially in the presence of sarcopenia) to improve response rates and outcomes [52,53].

Among the strengths of our study is having used two comparable cohorts (with an adequate size), homogeneous in characteristics (only HGSOC in the advanced stage), and different ways of evaluating the role of body composition and metabolic disorders in survival in HGSOC, along with sufficient follow-up data. One of the limitations of our study is the retrospective nature of our series, where nutritional interventions and the use of statins were not a routine indication and continuity or adherence to them over time was not evaluated. Our discoveries warrant a prospective study where the duration, treatment schemes, adherence to recommendations, and other interacting therapies are evaluated in addition to the primary survival outcomes.

## 5. Conclusions

Based on our findings, we believe it is relevant to propose the abandonment of BMI as an element of judgment to define or limit therapies at the time of diagnosis or recurrence. Incorporating more precise parameters, measurable by a CT scan, that enable a better definition of body composition can allow more effective interventions and eventually positively condition the response to therapeutic alternatives in this cancer, including immunotherapy.

More importantly, frontline therapeutic actions should include nutritional interventions (i.e., lifestyle and dietary changes) that tend to normalize body composition in the mid-term and correct lipid metabolism disorders (e.g., the prescription of statins) [54,55]. These interventions can not only contribute to improving therapeutic outcomes but also undoubtedly improve the quality of life for the patients, even in recurrence [56]. Thus, these interventions deserve, beyond any doubt, further research.

## Figures and Tables

**Figure 1 cancers-15-01156-f001:**
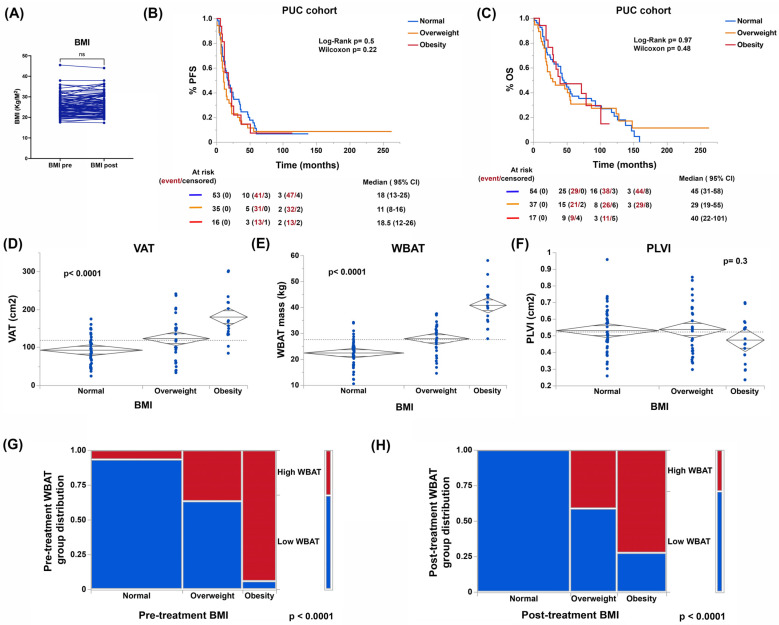
BMI trends (**A**), its impact on progression-free (PFS, **B**) and overall survival (OS, **C**), and its correlation with adiposity (visceral adiposity [VAT], whole body adipose tissue [WBAT]) and muscle mass (psoas to lumbar vertebrae index [PLVI]) (**D**–**F**) pre- and post- completion of planned treatment (**G**,**H**) in the PUC cohort. NS stands for non-significant.

**Figure 2 cancers-15-01156-f002:**
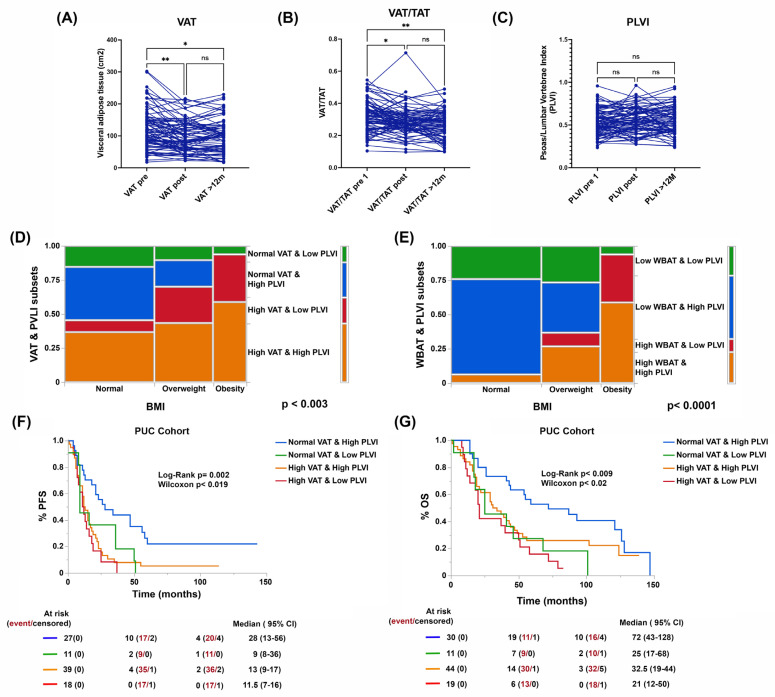
Trends in visceral adiposity (visceral to total adipose tissue [TAT] at the same level, **A**,**B**) and PLVI (**C**) and prevalence of different body compositions (BCs) defined by VAT or WBAT and PLVI after front-line treatment (**D**,**E**) and its impact in PFS and OS (**F**,**G**) in the PUC cohort. The * and ** indicate statistical significance among the groups (*p* < 0.001). ns stands for non-significant.

**Figure 3 cancers-15-01156-f003:**
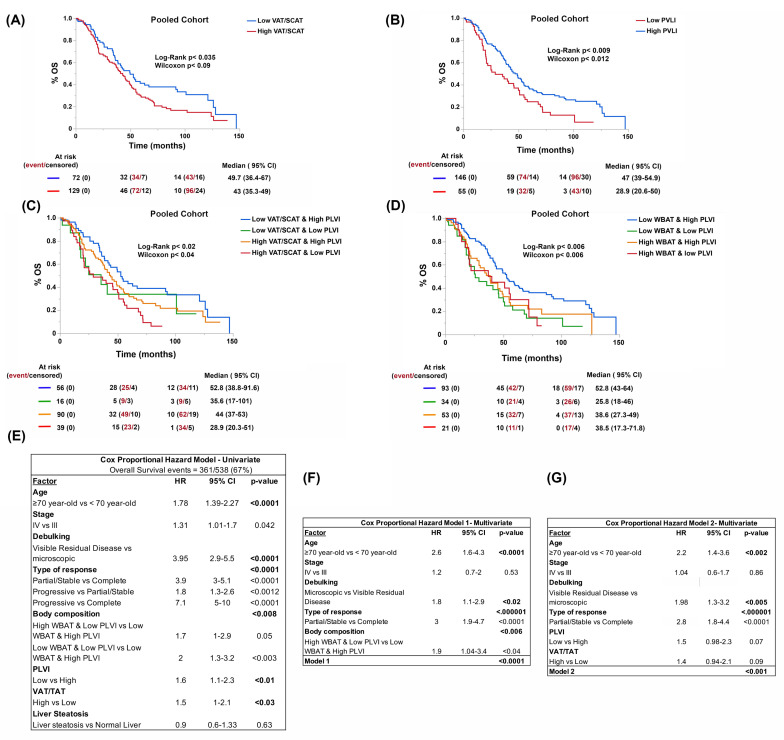
Overall survival (OS) curves depending on visceral to subcutaneous adipose tissue ratio (SCAT at same level) (**A**), PLVI (**B**), and BCs defined by VAT/SCAT (**C**) or WBAT (**D**) and PLVI, respectively. Univariate Cox regression analyses (**E**) and multivariate Cox-regression models (**F**) built up with prognostic factors (**E**–**G**) in the pooled cohort.

**Figure 4 cancers-15-01156-f004:**
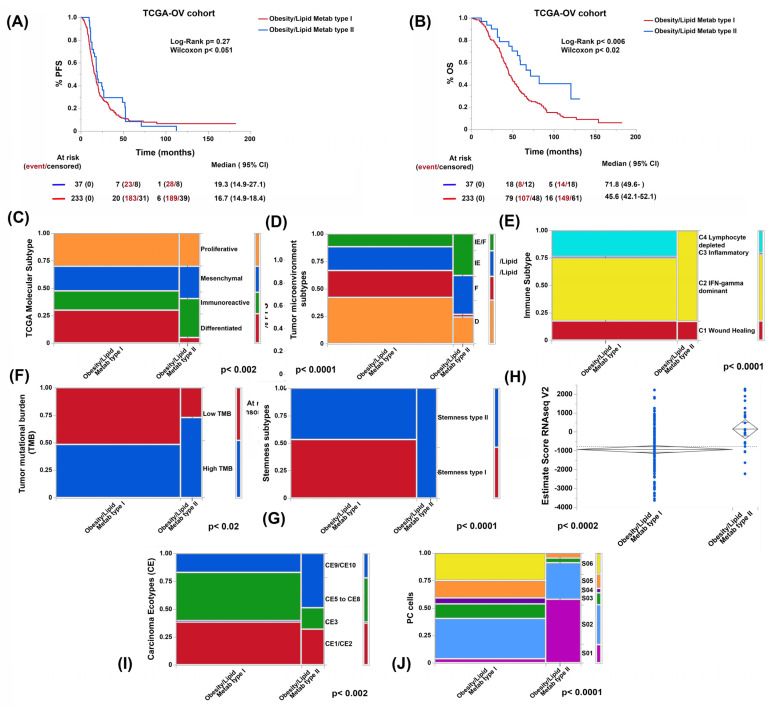
Comparative progression-free (PFS; **A**) and overall survival (OS; **B**) curves, prevalence of different TCGA-OV molecular subtypes (**C**), tumor microenvironment (TME) subtypes (desert [D], fibrotic [F], immune-enriched non-fibrotic [IE], and immune-enriched fibrotic [IE/F]; (**D**), immune subtypes (**E**), tumor mutation burden level (TMB; **F**), stemness subtype (**G**), ESTIMATE score (**H**), and prevalence of carcinoma ecotypes (CE, **I**) and plasma cell (PC) states (S, **J**) according to obesity and lipid metabolism clustering using RNA-seq data from the TCGA-OV cohort.

**Figure 5 cancers-15-01156-f005:**
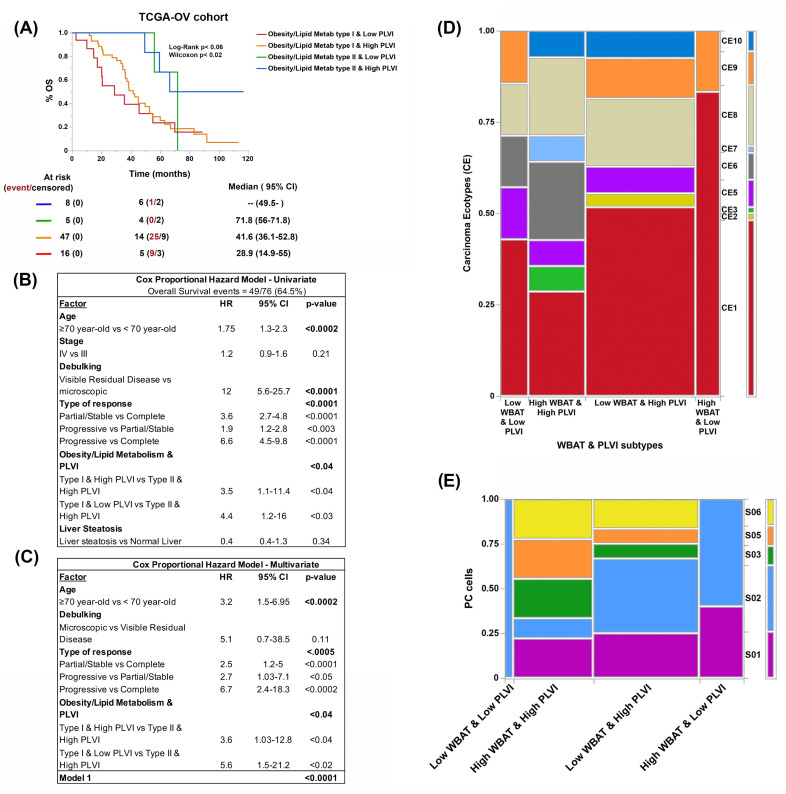
Overall survival (OS, **A**) curve and univariate (**B**) and multivariate (**C**) Cox proportional hazard regression analyses based on BC estimated using obesity and lipid metabolism clustering and central sarcopenia (PLVI, **A** and tables below it). Prevalence of different carcinoma ecotypes (CE) and plasma cell (PC) states (S) according to BC types (using VAT/SCAT and PLVI, **D**,**E**) in the TCGA-OV cohort.

**Figure 6 cancers-15-01156-f006:**
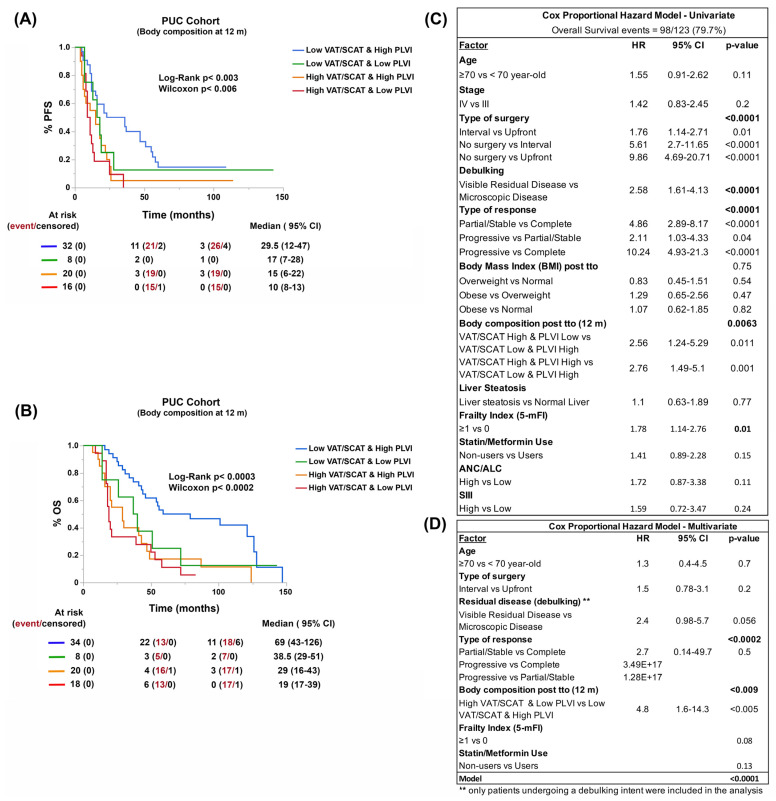
Effects of body composition (BC) changes (estimated by VAT/SCAT and PLVI) after one-year treatment on progression-free (PFS, **A**) and overall survival (OS, **B**) in the **PUC cohort**. Univariate and multivariate Cox proportional regression analyses (**C**,**D**) carried out to assess BC as an independent risk factor.

**Table 1 cancers-15-01156-t001:** Comparative clinical characteristics of the PUC and TCGA-OV cohorts.

Variable	PUC Cohort	TCGA-OV Cohort	*p*-Value
**n sample** (n with imaging)	123 (104)	n = 415 (97)	
**Age** (mean ± SD)	59 ± 11.6	59 ± 11.5	0.98
(min–max; range)	(29–83; 54)	(26–87; 61)	
**Stage**			0.67
III (%)	105 (85.4)	345 (83.1)	
IV (%)	18 (14.6)	70 (16.9)	
**Adiposity Estimates** VAT (cm^2^)	118.1 ± 58.9	130.8 ± 70	0.91
VAT/SCAT	0.49 ± 0.22	0.50 ± 0.22	0.55
VAT/TAT	0.32 ± 0.09	0.32 ± 0.09	0.55
WBAT (kg)	27.3 ± 9.2	30 ± 11.7	0.96
**Liver Steatosis** (CTl-s)			0.015
Normal	85 (81.9)	65 (67)	
Steatosis	19 (18.1)	32 (33)	
**Type of Surgery ***			
Upfront surgery	63 (51.2)	414 (99.8)	
Interval surgery	46 (37.4)	1 (0.2)	
Never surgery #	14 (11.4)	N/A	
**Residual Disease ***			<0.0001
Microscopic	55 (44.7)	86(20.7)	
Else residual disease	68 (55.3)	329 (79.3)	
**Type of Response**			0.52
Complete	76 (61.8)	286 (68.9)	
Partial/Stable	35 (28.5)	90 (21.7)	
Progressive	12 (9.8)	39 (9.4)	
Statin/Metformin Use	31 (25.2)	N/A	
**Mean/Median FU** (months)	58.9/44	44.5/38.3	<0.0001
(min-max; range)	(1–263; 262)	(1–183; 182)	
**Deaths** (%)	98 (79.7)	263 (63.4)	0.0005

N/A (not available). * Only patients with information on debulking status (reported in the surgical protocol) or a residual tumor (assessed by CT imaging after completion of treatment) were included. **#** inoperable cases based on extensive disease and/or severe co-morbidities.

## Data Availability

Raw counts, upper-quartile normalized fragments per kilo base per million mapped (FPKM-UQ) RNA-seq expression, and clinical data can be accessed and downloaded from Genomic Data Commons through the TCGA-biolinks R package.

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
