# Peer review of "Body Composition and Metabolic Dysfunction Really Matter for the Achievement of Better Outcomes in High-Grade Serous Ovarian Cancer"

_cancers, 2023, doi:10.3390/cancers15041156_

Round 1
Reviewer 1 Report
Well done publication.
Author Response
Dear reviewer, first, thank you for your interest in our manuscript and for your excellent opinion about it and your positive evaluation in favor of its publication. Regardless of this, we would like to tell you that we have introduced some changes specifically in response to the opinion of the other referees. Such changes, we believe, help to improve the original version by detailing some statistical methodological aspects, expressing some terms differently, and modifying some figures and tables to facilitate reading for interested readers.
Reviewer 2 Report
Cuello et al provide an interesting description of aspects related to body composition and metabolic disorders in women with HGSOC.
The article is well-written and the results are presented clearly.
I have minor suggestions:
1. Exclude lines 97-100 from the introduction as they form part of the results section.
2. Explain better how the covariates of the Cox models were defined.
3. Check if the VAT/SCAT and VAT/TAT values in table 1 are correct.
Author Response
Dear Reviewer, first, we want to thank your comments and positive evaluation of our manuscript. Second, we addressed all your observations that no doubt improved the original version of our manuscript.
1) Exclude lines 97-100 from the introduction as they form part of the results section.
Thanks for noticing such a redundancy. As recommended, we have modified and excluded such a paragraph from the revised version
- Explain better how the covariates of the Cox models were defined.
Thanks for highlighting this methodological aspect. We have better explained in the method section that we used stepwise method with a stopping rule or selection criteria based on p-value (0.2) to identify covariates to be included in Cox models. We have also added a couple of references for those readers interested in accessing further detailed information about recommendations on variable selection when interested in clinical prediction modelling.
- Check if the VAT/SCAT and VAT/TAT values in table 1 are correct.
Once again thanks for your insightful reading of our manuscript and for detecting potential mistakes in building up such a table. We have checked our database, individual numbers, p-values of each variable included there. As you will see we have corrected and modified the original table not only to correct any potential typo error made but also to add some variables recommended for one of the other referees. Specifically, VAT/SCAT and VAT/TAT were right. Their values become equal after rounding numbers. In fact, mean values for VAT/TAT were 0.317085 and 0.318840 for PUC and TCG-OV cohorts, respectively. Their respective SD were 0.0921757 and 0.0902774. For VAT/SCAT mean values ± SD were 0.4927128 ± 0.2161998 and 0.496929 ± 0.2243334, respectively.
Reviewer 3 Report
Dear Authors,
Please find my comments in the attached pdf. Overall the paper is well written and clear in it's parts. Some issues regarding the performed analyses and data presentation should be dealt with.
Hope you'll find my suggestions useful.

Author Response
Dear reviewer, we want to deeply thank you for your detailed, insightful, and wise observations. Based on your suggestions, we believed our current revised manuscript version has significantly improved in quality and reader experience compared with our original submission. We have addressed each of your observations as indicated in the pdf version that you attached in your review. We hope, we have answered all of them.
Comment in Line 30: Uppercase
As requested, we have used uppercase letter at the beginning of sentence.
Comment to sentence in Lines 52 to 54: best case scenario is complete resection at up-front surgery, then complete resection after neo adjuvant therapy. having residual disease, even minimum, is always worse. Complete surgery is the most important factor. Please rephrase this sentence to highlight this aspect. see this paper too http:// dx. doi. org/ 10. 1136/ ijgc-2020- 001658
As requested, we have re-phrased this sentence to highlight the relevance of upfront optimal surgical debulking. We also read the paper recommended and we decided to add as relevant reference.
Comment to Line 90: excellent! this is an important factor to consider. Always
Completely, we agree. Thanks for the comment.
Comment to Line 107: patient maybe sounds better
As recommended, we have changed to patients
Comment in Line 111: did any of your patients have neo-adjuvant chemotherapy? this is important if you want your cohort to be similar to that of the TCGA where all patients are up-front (if I remember well)
Thanks for this insightful comment.
As you comment, the complete TCGA-OV cohort was mainly conformed by advanced stage HGSOC cases who had an upfront debulking intent (482 patients [82.4%] of TCGA OV Pan Cancer atlas subset). From this cohort, 102 cases were discarded because no information was available in terms of surgical approach or debulking status (independently if all had been surgically approached upfront). Our cohort had a proportion of interval debulking cases as well as cases who never underwent surgery. From this cohort we also excluded 22 cases for similar reasons as we did with TCGA-OV cohort. We have included such an information in text (patient data collection, line 116) and in the new version of table 1. We also want to highlight that we carried out analyses in the pooled cohort in the intent of analyzing cases undergoing interval debulking surgery as well as those never operated on. If excluded, those cases from either our cohort or from the pooled cohort, results did not change significantly. As an example, we attached a table for your review where we compared the original PUC cohort with the subset where only upfront cases are included.
|
|
PUC complete cohort |
Upfront surgery subset |
|
p-value |
|
|
|
n=123 |
|
n=63 |
|
|
|
Age (mean ± SD) |
59 ± 11.6 |
|
55.65 ± 10 |
|
0.06 |
|
(min-max; range) |
(29-83; 54) |
|
28.8-80 |
|
|
|
Stage |
|
|
|
|
|
|
III (%) |
105 (85.4) |
|
56 (88.9) |
|
NS |
|
IV (%) |
18 (14.6) |
|
7 (11.1) |
|
|
|
Adiposity estimates |
|
|
|
|
|
|
VAT (cm2) |
118.1 ± 58.9 |
|
118.7 ± 62.7 |
|
NS |
|
VAT/SCAT |
0.49 ± 0.2 |
|
0.49 ± 0.23 |
|
NS |
|
VAT/TAT |
0.32 ± 0.09 |
|
0.32 ± 0.1 |
|
NS |
|
WBAT (kg) |
27.3 ± 9.2 |
|
27.06 ± 8.1 |
|
NS |
|
Liver Steatosis (CTl-s) |
|
|
|
|
|
|
Normal |
85 (81.9) |
|
40 (85.1) |
|
NS |
|
Steatosis |
19 (18.1) |
|
7 (14.9) |
|
NS |
|
Debulking* |
|
|
|
|
|
|
Microscopic |
55 (44.7) |
|
37 (58.7) |
|
0.083 |
|
Else residual disease |
68 (55.3) |
|
26 (41.3) |
|
|
|
Type of Response |
|
|
|
|
0.083 |
|
Complete |
76 (61.8) |
|
48 (76.2) |
|
|
|
Partial/stable |
35 (28.5) |
|
13 (20.6) |
|
|
|
Progressive |
12 (9.8) |
|
2 (3.2) |
|
|
|
Statin/Metformin Use |
31 (25.2) |
|
18 (28.6) |
|
NS |
|
Mean/Median FU (months) |
58.9/44 |
|
73.1/72 |
|
0.06 |
|
(min-max; range) |
(1-263; 262) |
|
(2-191; 189) |
|
|
|
Deaths (%) |
98 (79.7) |
|
46 (73.02) |
|
NS |
Comment in Line 116: I believe you should mention the approval from the TCGA
As requested, we have added to the respective sentence ‘publicly available’ in line 123. We also acknowledged at the end of manuscript the open access to TCGA datasets granted through the GDC data portal.
Comment in line 162 Please detail also by adding the results regarding the univariable analysis, how you selected the variables to include in the multivariable analysis and in the results present these findings a bit more. So comment on which group is at rick with that HR and even better the % increase in risk, how statin use remains an independent factor etc.
general workflow woould be like this:
perform univariable analysis on all variables.
select the variables for the multivar analysis (the decision on which variable to include depend on you. the most commont approaches are: insert in the multivar only the significant univar variables, insert the variables with p<0.1 or p<0.2 or insert the significant var+those clinically relevant in literature whichever the p value in the univariate analysis)
do the multivar and present the results in the text
If needed I would advise to seed help from an experienced statistician
As requested, we have detailed how variable selection was made and what method was used to carry out Cox models. Effectively, we first carried out univariate analyses for all variables. As you commented, we indeed used a stepwise approach including all those variables with significant p-value in univariate analyses. To select variables, we used a stopping rule or selection criteria based on p-value 0.2. We also added variable based on clinical relevance (e.g., molecular subtype, sensitivity to first line treatment, stage) already reported in the literature (we added a couple of references about it), independently of p-value found. We have added a reference for those interested in reviewing methodological aspects. To further address your observation, we have modified the information contained in respective graphs (including the most relevant factors). In this corrected version, we added the univariate Cox analyses, the HR, and changed the HR data presentation to facilitate interpretation.
Comment in line 180 related to table 1: I would add the title "variable" to the first column. Also, if possible, I would left align column 1 with a little indentation for the subgroups of the variables so that it stands out easier. TCGA-OV.
As requested, we add the word ‘variable’ to the first column. In addition, we used bold letters to highlight variables in the modified version of the table 1. We also changed the TCGA denomination to TCGA-OV (we did the same whenever appearing).
Comments on Figure 1 in line 198: could you please change the order to normal overweight obesity? like this the order would be based on the severity so it would be more elegant. also please enlarge this figure as a whole. the fonts are small and since there's no space limitation, one might as well take advantage of it.
for KM curves: please add censoring as well as risk tables under the curves. Something more like the size of Fig S5 for example.
which? surgery?
As requested, we have changed the order of variables in the graph following your suggestions. We have enlarged the whole figure and letter size. As you will see, in all figures we have added the at-risk tables at different time points under survival curves.
In relation with the question of which treatment? We modified to ‘completion of planned treatment’ meaning after surgery plus chemotherapy
Comments in line 203 referred to Figure 2 and others: same comment regarding the size and the KM curves. Avoid dotted lines in KM curves. cooment for all figures: abbrev must be written in full so that figures are self-explanatory.
when performing multiple comparisons, whch curves are most divverent from the others? Also if possible, I would add the median PFS ans OS with 95%CI in a table under the curves (or wherever you find it more appropriate)
As recommended, we have change dotted line for other colored lines. To solve the abbreviation issue, we have included a description of each one in the figure legend to avoid overfill the graph’s sight of the reader. As required by the journal, figures appears immediately before or after first mentioned in text. As you could see, abbreviations have main already fully explained before its first appearance in each graph.
With respect of what curves are more divergent or different (there is a typo mistake in your comment)? We have added a sentence highlighting such a difference among comparison. We also have added the median FU with 95% CI for each group both in PFS and OS.
Comment in line 228: was this done by AUC analysis or just by choosing the quartile?
With respect to the cutoff used with PLVI to define sarcopenia, as described in methodology and in result section, we chose to use the lowest quartile in each cohort as well as in the pooled cohort rather than finding a cutoff based on AUC. Such a cutoff is more than one standard deviation from the mean of each cohort. According to other publications, one cited in the manuscript (reference 27), the mean value of PLVI found among women without cancer is 0.77 ± 0.21 at similar ages. Definition of Sarcopenia, and stratification of High and Low PLVI in this and other papers have been based on the cohort median and percentiles.
Comment in line 228 referring to Figure 3: add HR not only the C. Also, it's CI not IC. also add both the univariate and the multivariante analysis so that one can see the change in p values and HRs. I find it impossible that a PR and SD can have a better outcome than a CR. please check you reference groups for all variables. It seems to me that there is a mistake here. To be clear I would advise to prepare the extended version of the table which leaves no doubts. Meaning that you have more rows:
RECIST
CR_____ref
PR_____HR xxx
SD_____HR xxx
same comment for all the COX analyses in the rest of the paper
As requested, we added the HR, we correct the typo error on CI (not IC) and included the univariate and multivariate analyses. We also have extended the tables with additional rows and comparisons.
Comment in line 243: in the TCGA-OV cohort
As requested, we have modified throughout the manuscript from TCGA to TCGA-OV and we clarify the title to which you referred.
Comment in line 246: in-house precompiled
Thanks for suggesting this change. We have added ‘in-house precompiled’
Comment in line 261 referring to Figure 4: PFS based on these 2 classes?
As requested, we have added the PFS curve to the figure. As you can see, there was a trend to better PFS in one of the classes. We have also commented on it in the specific section of the manuscript.
Comment in line 307 and 308 referred to figure 5: this should be G.
How does BMI stand in this analysis?
If you could fing a way to put the TCGA or PUC label in the figures it would be great. For example, like in FigS4
As recommended, we have named univariate and multivariate analyses as (G) in the respective figure.
We also thanks your insightful comment on BMI. We have included in tables to highlight its null role.
With respect to entitling properly graphs. We added TCGA-OV or PUC or Pooled cohort to facilitate readers’ understanding.
Comment in line 312: this info should be detailed abotve when you present the cohorts of patients.
As advised, we have first mentioned and included statin/metformin data in table 1 when introducing the cohorts.
Finally, I want to thank once again for all your comments and observations. All give us the opportunity to improve our research and current manuscript.
Round 2
Reviewer 3 Report
Dear Authors,
Thank you for the time invested in answering my comments (indeed there were some typos. sorry about that. I hope they didn't give you too much trouble)
Last remarks:
- in the captions where there are KM curves, mention that the risk tables have events and censored patients in parentheses (if this is what that number means).
- please align tables in figure 3
This aspect is still unclear to me: for what I understand, your cohort has patients with and without surgery. In the COX analysis you have the residual disease after debulking, thus the COX analysis has been conducted only on patients undergoing primary debulking? so a subgroup of the whole cohort? Also, In table 1 you have 123 patients for which you report residual disease after debulking for all 123. So where are the inoperable patients? this is really unclear... Please clarify the inclusion and exclusion criteria in material and methods section.
For what I understand, in the end you included only patients undergoing surgery, so, please present in table 1 how many patients had primary and interval surgery. Also, this should be considered in the COX analysis since it's widely shown that it significantly impacts patient outcomes.
Thank you!
Author Response
Dear reviewer, once again we want to thank you for your dedicated and thorough reviewing work. You are right with the observations. Here, we clarify your last remarks point by point, as following:
Last remarks:
1) in the captions where there are KM curves, mention that the risk tables have events and censored patients in parentheses (if this is what that number means).
As requested, we have added the events besides the censored cases. To facilitate the understanding of reader, we differentiate them using color and slash within parenthesis. In each line you can see the At risk patients and within parenthesis (events/ censored).
2) please align tables in figure 3
As advised, we have change the disposition of table 3, aligning them.
3) This aspect is still unclear to me: for what I understand, your cohort has patients with and without surgery. In the COX analysis you have the residual disease after debulking, thus the COX analysis has been conducted only on patients undergoing primary debulking? so a subgroup of the whole cohort? Also, In table 1 you have 123 patients for which you report residual disease after debulking for all 123. So where are the inoperable patients? this is really unclear... Please clarify the inclusion and exclusion criteria in material and methods section.
Once again, thanks for asking this clarification. We have specified the inclusion and exclusion criteria in material and methods section. As you will see in the respective paragraph (line 105) we modified sentences to specifically explain not only inclusion but also exclusion criteria.
Specifically answering about what patients were included in multivariate Cox analyses, we did both manners in PUC cohort since information was available in all cases collected by us. For analyses carried out with the Pooled cohort or the TCGA-OV cohort, we only included cases undergoing an intent of debulking (either upfront or as interval surgery). Then, as you insightfully questioned, we clarified that analyses made in these cohorts reflecting results of a surgical-only subgroup (upfront or interval debulking surgery). We specified the point in the respective section results.
About table 1, we changed from debulking to residual disease after completion of treatment as stated either by the surgical report (for those cases undergoing debulking intent) or by CT imaging (for those cases never operated on). Our fourteen cases, never undergoing surgery, are included in the 'Else residual disease group' and correspond mainly to inoperable cases due to extensive disease or severe co-morbidities. We added a footnote to specify this point.
3) For what I understand, in the end you included only patients undergoing surgery, so, please present in table 1 how many patients had primary and interval surgery. Also, this should be considered in the COX analysis since it's widely shown that it significantly impacts patient outcomes.
To clarify this point, we detail more table 1. Now, patients undergoing surgery (either upfront or interval) and those patients never operated on are clearly identified. To mention, such information was available only in our PUC cohort but not in TCGA-OV. Unluckily, the TCGA-OV included cases where information was 'incomplete' or 'unclear'. As replied previously, TCGA-OV was composed mainly for upfront surgically approached cases but not exclusively. In fact the cohort collected here includes one case of interval surgery. For your knowledge, those cases denominated by us as information 'incomplete/unclear' correspond to cases diagnosed by 'cytology/biopsy' as reported in the downloadable clinical database without information about tumor burden, if they really underwent surgery or not, and lacking on reliable estimation of residual tumor after treatment. Obviously, those TCGA-OV cases were excluded from analyses. In the modified table 1, such missing information appears now as N/A (not available) in the corresponding row. Finally, about including primary vs interval surgery in Cox model, such variable was added in multivariate analyses in PUC cohort since numbers allowed it. We added such a information in respective tables. We agree with its relevance and specified that cases included here correspond to cases undergoing interval surgery in timely planned fashion and not delayed surgeries (PUC cohort since we could register such a variable). You will see reflected in the new version of the multivariate Cox analyses of such a cohort that type of surgery was specifically added on your request (Figure 6 D). Such a inclusion did not modify significantly the complete model.
Round 3
Reviewer 3 Report
Dear Authors,
Thank you for your time and clarifications. I appreciate your efforts.
Best of luck with your future research!